# Stimuli-Responsive Designer Supramolecular Polymer Gel

**M. Douzapau, Srayoshi Roy Chowdhury, Surajit Singh, Olamilekan Joseph Ibukun and Debasish Haldar ***

Department of Chemical Sciences, Indian Institute of Science Education and Research Kolkata, Mohanpur 741246, West Bengal, India

* Correspondence: deba_h76@yahoo.com or deba_h76@iiserkol.ac.in

**Abstract:** This paper reports a stimuli-responsive designer supramolecular polymer gel in dimethyl-sulphoxide (DMSO)/water (1:2) based on a dipeptide amphiphile and β-cyclodextrin (β-CD) The dipeptide amphiphile contains caproic acid at the N terminus and methyl ester at the C terminus. From X-ray single crystal diffraction, the amphiphile adopts a kink-like conformation. The amphiphile self-assembled to form a parallel sheet-like structure stabilized by multiple intermolecular hydrogen bonds. Moreover, the parallel sheet-like structure is also stabilized by edge-to-edge π–π stacking interactions. In higher-order packing, it forms a corrugated sheet-like structure stabilized by hydrophobic interactions. The dipeptide amphiphile interacts with β-cyclodextrin and forms gel through supramolecular polymer formation in (DMSO)/water (1:2) by a simple heating-cooling cycle. The sol-to-gel transformation is because of a host–guest complex between compound **1** and β-CD and the formation of supramolecular polymer accompanied by microstructure changes from nanofibers to microrods. The gel is temperature responsive with a $T_{gel}$ of 70 °C. The supramolecular polymer gel is also responsive to stimuli such aspicric acid and HCl. The extensive spectroscopic studies show that the aromatic hydrophobic side chain of compound **1** forms a host–guest complex with β-CD. These results will be helpful for the design of advanced programable eco-friendly functional materials.

**Keywords:** supramolecular; programmable; stimulus-responsive; host–guest complex; crystal structure; gel





## 1. Introduction

Supramolecular polymers are the directed assembly of designer building blocks that are cross-linked by non-covalent interactions [1–5]. The recognition and assembly of thesedesigner building blocks by various noncovalent interactions such as hydrogen bonding, π–π stacking, ion-pair interactions, van der Wall's interactions, and hydrophobic interactions are the key to developing the supramolecular polymer [5]. Due to these reversible non-covalent interactions, supramolecular polymers are degradable, recyclable, and show self-healing propensities [6–9]. Wang and co-workers reported robustsupramolecular elastomers synergistically cross-linked by noncovalent interactions such as hydrogen bonds and coordination bonds, which exhibit properties including high tensile stress, exceptional toughness, high stretchability, and excellent self-healing ability [10]. The supramolecular polymers and gels are stimuli-responsive functional materials [11] and are valuable commodities for their versatile applications, including in the construction of inter alia self-healing materials, memory retention materials, drug delivery systems, and extracellular matrix for tissue engineering and regeneration [10–20]. Yan and coworkers developed cross-linked supramolecular polymeric materials as substrates for stretchable, anti-tearing, and self-healable thin film electrodes [21]. Moreover, changing the building blocks used to produce the supramolecular polymer—which creates new systems with different properties such as thiophene- or acene-based supramolecular polymers—can be applied in organic photovoltaic devices [22].Building blocks containing gas-sensitive functional groups, such as o-phenylenediamine (NO-sensitive), diethylamine ($CO_2$-sensitive),

and o-azidomethylbenzoate ($H_2S$-sensitive) can be used for the development of gas-responsive supramolecular material [23]. Photo-sensitive functional groups such as azobenzene [24], anthracene [25], and spirobenzopyran [26,27] formed supramolecular polymers with photo-responsive applications [28–30]. Cyclodextrin is an important building block for supramolecular polymer formation by the host–guest inclusion complex [31]. Previously, Rekharsky and coworkers examined the complexation thermodynamics of water-cyclodextrins system and reported that α-CD and γ-CD are more favorable to forminga host–guest complex by alkyl chain than β-CD [32]. However, Bhattacharyya et al. reported that DMF molecules exhibit markedly slower solvation dynamics in β-CD than those for water in γ-CD because inside the cavity motion of the confined DMF molecules is highly constrained [33].

Herein, we report a stimuli-responsive supramolecular polymer gel developed by an amphiphile and β-cyclodextrin (β-CD) in DMSO/water (1:2). From X-ray single crystal diffraction, the amphiphile adopts a kink-like conformation. The amphiphile self-assembled to form a parallel sheet-like structure stabilized by multiple intermolecular hydrogen bonds. Moreover, the parallel sheet-like structure is also stabilized by edge-to-edge π–π stacking interactions. In higher-order packing, it forms a corrugated sheet-like structure stabilized by hydrophobic interactions. The amphiphile interacts with β-cyclodextrin and forms gel through supramolecular polymer formation in (DMSO)/water (1:2) by a simple heating-cooling cycle. The gel is temperature responsive and $T_{gel}$ is 70 °C. The sol-to-gel transformation is accompanied by microstructure changes from nanofibers to microrods. The supramolecular polymer gel is responsive to stimuli such as picric acid and hydrochloric acid. The absorption and emission spectroscopic studies show that the aromatic hydrophobic side chain forms an inclusion complex with β-CD. The stimuli-responsive designer supramolecular gel has the potential to be aneco-friendly advanced functional material.

## 2. Experimental

*Materials and Reagents*

Sigma Chemicals supplied D-phenylalanine, L-alanine, β-cyclodextrin, caproic acid, and DMSO. All chemicals were used without further purification.

Synthesis of Compound **1**

**Synthesis of D-PheOMe.HCl 2**: An amount of 6.6 g D-Phe (40 mmol) was placed in a dry round bottom flask, and absolute methanol (10 mL) was quickly added and closed with a $CaCl_2$ guard tube. The $CaCl_2$ guard tube was used as a lid throughout the reaction to protect the content from moisture from the air. The solution was cooled in an icebath for 5 min. Thionyl chloride (5 mL) was cautiously added to the methanol solution and stirred for 12 h. The excess methanol and $SOCl_2$ were evaporated under reduced pressure to obtain D-PheOMe.HCl as a white solid and was used without further purification.

**Synthesis of caproic acid appended D-phenylalanine 3:** An amount of 1.8 g (15.18 mmol) of caproic acid was dissolved in dry dichloromethane (DCM) and kept in an ice bath and 4.1 g (20 mmol) of Phe-OMe.HCl (isolated from acid protection reaction of D-Phe by $SOCl_2$, MeOH) was added, followed by 4.12 g (20 mmol) dicyclohexylcarbodiimide (DCC) and 2.7 g hydroxybenzotriazole) HOBt and 4 mL $Et_3N$ (Scheme 1). The reaction mixture was allowed to sit until it reached room temperature and stirred for 2 days. Then, DCM was evaporated, and the residue was dissolved in ethyl acetate (60 mL). The organic layer was washed with 2 M HCl (3 × 50 mL) and 1 M $Na_2CO_3$ (3 × 50 mL), then dried over anhydrous sodium sulfate, and evaporated undervacuum to obtaina white solid. The yield was 75%.

**Scheme 1.** Synthesis of compound **1**.

**NMR spectroscopy.** [1]H NMR spectra of the final compound were recorded with a JEOL 400 MHz NMR spectrometer at 278 K for 512 scans. Compound concentrations were in the range of 1–10 mM in $CDCl_3$ and TMS as an internal standard.

**FT-IR spectroscopy.** All reported solid-state FTIR spectra were obtained with a Perkin Elmer Spectrum RX1 spectrophotometer with the KBr disk technique.

**ESI-MS spectrometry.** ESI-MS spectra were recorded on a Q-Tof Micro YA263 high-resolution (Waters Corporation) mass spectrometer by positive-mode electrospray ionization.

**Saponification ofcaproic acid appended D-phenylalanine 4**: An amount of 16 mmol of the above product was added in 25 mL MeOH and 2(M) 12.5 mL NaOH and stirred for 12 h. Then, methanol was evaporated, and the residue was dissolved in 50 mL of water (Scheme 1). Diethyl ether (2 × 50 mL) was used to wash the aqueous layer. Then, the pH of the solution was adjusted by adding 1(M) HCl and the compound was extracted with ethyl acetate (3 × 50 mL). The extracts were dried over anhydrous sodium sulfateand evaporated under vacuum to obtain a white solid. Yield is 85%.

**Compound 1:** An amount of 10 mmol of compound **3** was dissolved in dry DCM and kept in an ice bath and 2.15 g (10 mmol) of Ala-OMe.HCl (isolated from acid protection reaction of L-Ala by $SOCl_2$, MeOH) was added, followed by 2.06 g (10 mmol) DCC, 1.35 g HOBt, and 2 mL $Et_3N$ (Scheme 1). The reaction mixture was allowed to sit until it reached room temperature and stirred for 2 days. Then, DCM was evaporated, and the residue was dissolved in ethyl acetate (60 mL). The dicyclohexyl urea (DCU) was filtered off. The organic layer was washed with 2 M HCl (3 × 50 mL) and 1 M $Na_2CO_3$ (3 × 50 mL), then dried over anhydrous sodium sulfate and evaporated under vacuum to obtain a yellowish-white solid. Purification was performed by a silica gel column (100–120 mesh size) with a 1:9 ethyl acetate and hexane mixture as the eluent. The compound was characterized by [1]H NMR, [13]C NMR, FT-IR spectroscopy, and mass spectrometry. The yield was 2.74 gm (7.1 mmol) 70%.

[1]H NMR (400 MHz, DMSO-$D_6$, δ ppm):8.45–8.44 [1H, d, NH], 7.98–7.97 [1H, d, NH], 7.26–7.18 [5H, m, ArH], 4,59–4.53 [1H,m, Phe CαH], 4.32–4.24 [1H, m, Ala CαH], 3.63 [3H, s, OMe-H], 3.03–2.99 [1H, m, Phe-CβH], 2.74–2.69 [1H, m, Phe-CβH], 2.01–1.98 [2H, t, Aliphatic-H], 1.37–1.30 [4H, m, Aliphatic-H].1.22–1.16 [3H, m, Ala-CβH], 1.08–1.03 [2H, m, Aliphatic-H], 0.82–0.79 [3H, t, Aliphatic-H].[13]C NMR (100 MHz, $CDCl_3$, δ ppm): 170.04, 170.3, 170.1, 130.7, 120.9, 120.8, 120.7, 50.4, 50.3, 40.8, 30.9, 30.1, 20.6, 20.2, 10.8, 10.4.

**FT-IR spectroscopy.** All reported solid-state FTIR spectra were obtained with a Perkin Elmer Spectrum RX1 spectrophotometer with the KBr disk technique.

**ESI-MS spectrometry.** ESI-MS spectra were recorded on a Q-Tof Micro YA263 high-resolution (Waters Corporation) mass spectrometer by positive-mode electrospray ionization.

**Polarized optical microscopy**: The morphology of the xerogels was determined by using polarized optical microscopy. A small gel was placed on a clean glass coverslip and then dried by slow evaporation at room temperature for 10 days and then visualized at $10\times$ magnification (Olympus optical microscope equipped with a polarizer and a CCD camera).

**Field emission scanning electron microscopy**: The morphologies of the xerogels were investigated using field emission scanning electron microscopy (FE-SEM). For FE-SEM, a small amount of gel was drop cast on a clean glass coverslip and was allowed to dry for 7 days by slow evaporation at room temperature. Finally, the sample was dried under reduced pressure for two days. The samples were gold-coated, and the micrographs were obtained using a Zeiss DSM 950 scanning electron microscope.

**Rheology**: An amount of 4 mg of compound **1** was dissolved in 200 µL DMSO and 400 µL of $H_2O$ using a 3 mL glass vial for rheology measurement. Then, the sample was heated at 70 °C for 2 min; after that, the sample was subjected to sonication for 30 s. After sonication, instant gelation was observed. The gel was kept at that state for 4 h before rheology measurement. All rheological measurements were undertaken on an Anton PaarPhysica MCR 102 rheometer at 25 °C. Strain, frequency, and temperature sweeps were performed using an 8 mm parallel plate geometry. Strain sweeps were performed at 10 rad/s from 0.01% to 100% strain. Frequency sweeps were carried out from 0.1 rad/s to 100 rad/s at 0.1% strain. All gels were left for ~4 h before being measured.

**Single crystal X-ray diffraction study**

Diffraction-quality crystals of compound **1** were obtained from the mixture of a solvent (methanol–water with a 3:1 ratio) by slow evaporation. Single-crystal X-ray analysis of compound **1** was carried out on a Rigaku Oxford Diffraction, 2015 X-ray diffractometer instrument with MoK$\alpha$ radiation. Data were processed using the CrysAlisPro 1.171.38.43f and the structure solution and refinement procedures were performed using ShelXL. Refinement of non-hydrogen atoms was performed using anisotropic thermal parameters. CCDC 2,243,507 contains the crystallographic data for compound **1**.

Crystallographic data of compound **1**: space group P 21, a = 11.1422(6), b = 4.8587(3), c = 17.6234(10) Å, $\alpha$ = 90°, $\beta$ = 96.473(6)°, $\gamma$ = 90°, V = 947.99(10) Å$^3$, Z = 2, dx = 1.221 gcm$^{-3}$, T = 100 K, R1 0.0501, and wR2 0.1262 for 3076 data with I > 2$\sigma$(I).

## 3. Results and Discussion

The amphiphile 1 (Figure 1) containing caproic acid at the N terminus, D-phenylalanine, L-alanine, and methyl ester at the C terminus was synthesized by solution reaction. The synthesized compound and all intermediates were purified and characterized by $^1$H NMR, $^{13}$C NMR, FT-IR spectroscopy, and ESI-MS spectrometry (Supporting Information). The motivation was to develop a compound that will form a supramolecular polymer and gel by including a complex with cyclodextrin. We tested all three macrocyclic oligosaccharides ($\alpha$-CD, $\beta$-CD, and $\gamma$-CD) (Figure 1). Cyclodextrin contains six/seven/eight glucose subunits covalently connected by $\alpha$-1,4 glycosidic bonds (Figure 1) [34]. The cyclodextrin has a toroid shape with a primary face (smaller opening) and a secondary face (larger opening). The 0.78 nm-thick toroid has a hydrophobic cavity diameter of 0.47 nm for $\alpha$-CD, 0.65 nm for $\beta$-CD, and 0.85 nm for $\gamma$-CD (Figure 1). The cavity inside cyclodextrin may host aliphatic or aromatic hydrocarbons and form an inclusion complex [35]. The exterior of cyclodextrin is hydrophilic, which helps to solubilize in water.

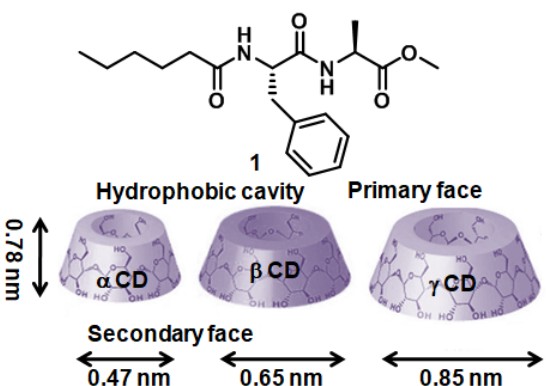

**Figure 1.** Schematic presentation of compound **1** and α, β, andγ-cyclodextrin.

Crystals of compound **1**, suitable for X-ray diffraction analysis, were developed from methanol–water solution by slow evaporation. The asymmetric unit has one molecule of compound **1**. Figure 2 shows the ORTEP diagram with the atom numbering scheme of compound **1**. From Figure 2, compound **1** adoptsa kink-like conformation. The backbone torsion angles are C6-N1-C7-C15 $\phi1 = 119.3°$; N1-C7-C15-N2 $\psi1 = -111.9°$ and C15-N2-C16-C18 $\phi2 = -145.2°$; N2-C16-C18-O4 $\psi2 = 174.6°$.

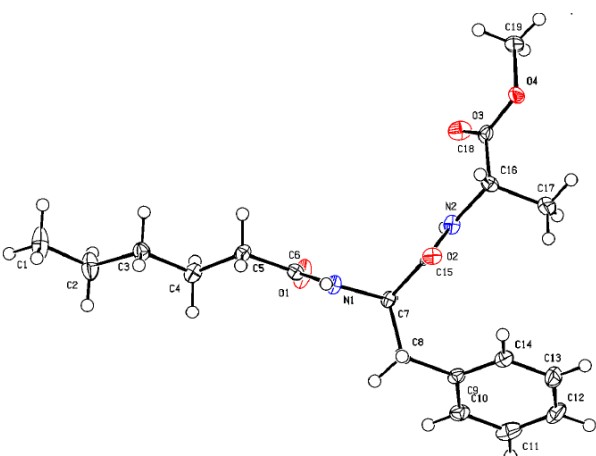

**Figure 2.** The ORTEP diagram with atom numbering scheme of compound **1**. 50% Probability. [C in grey, O in red, N in blue].

Further, compound **1** forms a parallel supramolecular sheet-like structure where four intermolecular N-H⋯O=C hydrogen bonds help to develop a robust structure (Figure 3a). Hydrogen bonding parameters for compound **1** are listed in Table 1. From X-ray crystallography, the compound **1** parallel sheet-like structure is also stabilized by $\pi$–$\pi$ stacking interactions (Figure 3b). The edge-to-edge $\pi$–$\pi$ stacking interaction is between the Phe aromatic rings where the centroid-to-centroid distance is 4.8Å, andthe shortest C-C distance is 3.7Å (Figure 3b). The crystal structure data analysis for $\pi$–$\pi$ interaction shows that the centroid–centroid distancecan vary between 3.4 Å to 5.6 Å [25,26].

**Table 1.** Hydrogen bonding parameters of compound **1**.

| D−H⋯A | D⋯H (Å) | H⋯A (Å) | D⋯A (Å) | D−H⋯A (°) |
|---|---|---|---|---|
| N1−H1⋯O1 [a] | 0.860 (3) | 2.112 (3) | 2.956 (3) | 166.9 (3) |
| N2−H2⋯O2 [b] | 0.860 (3) | 2.076 (3) | 2.903 (3) | 161.1 (3) |

Symmetry equivalent: a = x,1 + y,z; b = x,−1 + y,z.

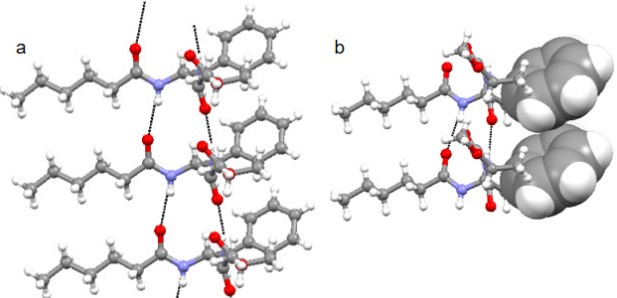

**Figure 3.** (**a**) The solid-state structure of compound **1** showing a parallel sheet-like structure stabilized by multiple intermolecular hydrogen bonds. Intermolecular hydrogen bonds are shown as black dotted lines. (**b**) Solid state structure of compound **1** showing edge-to-edge π–π stacking interaction. π–π stacking interactions are shown as a space fill model; [C in grey, O in red, N in blue].

The packing diagram shows that compound **1** was further assembled to form a supramolecular corrugated sheet-like structure along crystallographic *a* and *c* directions (Figure 4). The corrugated sheet-like structure was stabilized by dispersion forces between caproic acid alkyl chains (Figure 4). The crystal data and refinement statistics for compound **1** are listed in Supplementary Materials Table S1.

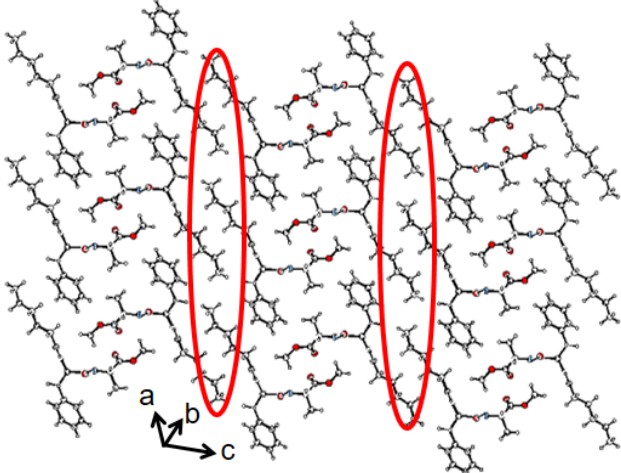

**Figure 4.** The solid-state packing diagram of compound **1** showing a supramolecular corrugated sheet-like structure stabilized by intermolecular hydrophobic interactions. Intermolecular hydrogen hydrophobic interactions are shown in red circles [C in grey, O in red, N in blue].

Compound **1** fails to form a gel in any solvent by heating-cooling cycles and sonication, even at high concentrations. Compound **1** contains hydrophobic groups (aliphatic and aromatic). Therefore, it may form the host–guest interactions with cyclodextrin and form supramolecular polymer and gel. Cyclodextrin is soluble only in water, DMF, and DMSO. We have tried with water and α-CD, β-CD, and γ-CD. Though compound **1** is insoluble in water, it forms a white suspension. Even after several heating-cooling cycles and sonication, it fails to form a gel. An amount of 4.0 mg of compound **1** and 25.0 mg of cyclodextrin (1:2 ratio) wereplaced in DMSO (200 μL) and $H_2O$ (400 μL) in a small vial, the vial was closed, heated, and sonicated for 30 s. After standing for 5 min, a yellow opaque gel was formed in β-CD. The gel formation was confirmed by the inverted vial method (Figure 5a). The host–guest inclusion complex stability depends on the cavity size of the CD and the size of the guest. The α-CD and γ-CD are more suitable forforming an inclusion complex with hydrophobic groups (alkyl chain) than βCD in water [27]. However, in this case, α-CD and γ-CD do not form a gel (Supplementary Materials Figure S1).

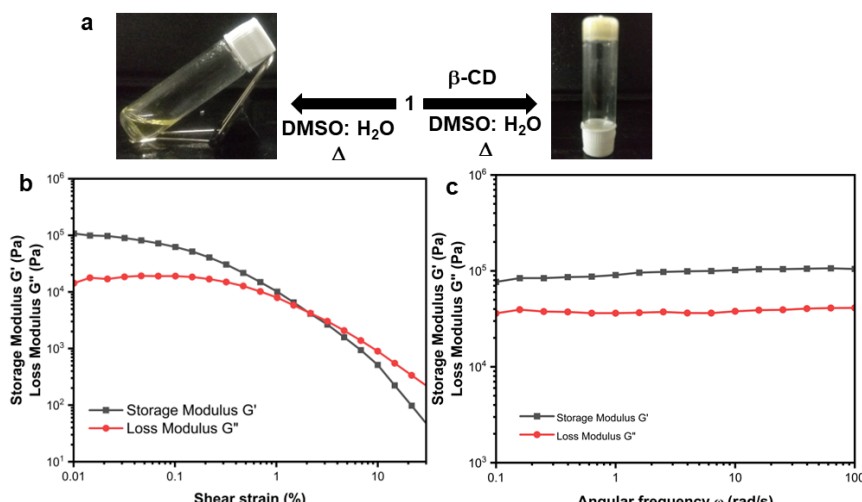

**Figure 5.** (**a**) The image showing compound **1** and β-CD supramolecular gel in DMSO-H$_2$O. (**b**) Effect of strain on supramolecular gel in DMSO-H$_2$O; (**c**) Effect of angular frequency on supramolecular gel in DMSO-H$_2$O.

Further, to examine the mechanical strength of the supramolecular gel, rheology experiments were performed. Both the angular frequency and oscillatory strain were measured. An Anton Paar Modular Compact Rheometer having a steel parallel plate geometry with an 8 mm diameter and 0.7 mm gap was used at 25 °C. The storage modulus G/(elastic response) and the loss modulus G//(viscous response) dissipated as heat. From the amplitude sweep experiment, the loss modulus G//becomes greater than the storage modulus G/at 2.3% oscillation strain, which indicates that the gel breaks above this strain (Figure 5b). G' > G" at 0.1% strain amplitude in the frequency sweep measurements. However, the storage modulus G/is two orders of magnitude higher than the loss modulus over the entire angular frequency range (Figure 5c). Hence, the gel is due to physical crosslink and is elastic in nature.

To investigate the morphological changes in supramolecular polymer and gel formation, xerogel was studied by field emission scanning electron microscopy (FE-SEM). The FE-SEM image of only compound **1** in DMSO depicted the morphology of the entangled nanofibers (Figure 6a). The nanofibers have a diameter of about 300 nm and are several micrometers in length. However, the microstructure of compound **1** and β-CD supramolecular gel in DMSO-H$_2$O (Figure 6b) exhibited the formation of polydisperse microrods with a length of about 30 μm and diameter ca 2 μm.

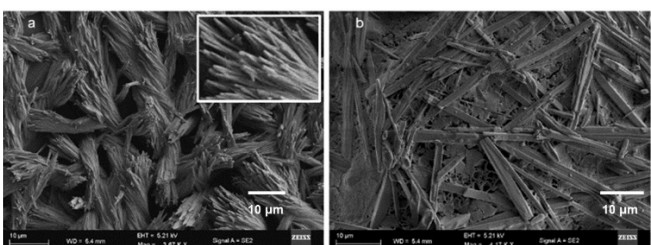

**Figure 6.** (**a**) FE-SEM image of compound **1** in DMSO and (**b**) FE-SEM image of xerogel form by compound **1**, β-CD in DMSO and water mixture.

The interaction of compound **1** and β-CD was studied by absorption and emission spectroscopy. With the addition of β-CD to 20 μM compound **1** in DMSO, the intensities of the absorption band at 280 nm increase. Further, we studied the host–guest complex formation of compound **1** and β-CD in DMSO by emission spectroscopy. The typical emission spectra of compound **1** show bands at 350 and 408 nm on excitation at 360 nm

(Figure 7a). With the gradual addition of β-CD, the emission intensities of the bands gradually increase (Figure 7a).

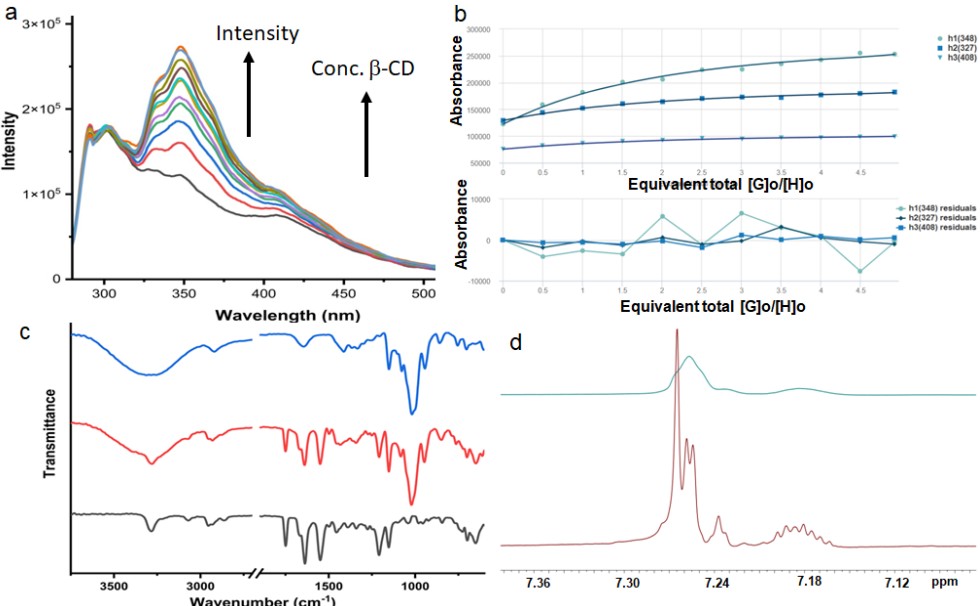

**Figure 7.** (**a**) The emission spectra of compound **1** show bands at 348 and 408 nm. The emission intensities of the bandsgradually increase with the gradual addition of β-CD (excitation at 280 nm). (**b**) The binding stoichiometry of compound 1 and β-CD in DMSO by fitting emission data at 348, 327, and 408 nm using the bindfit methods, http://supramolecular.org (accessed on 21 February 2023). (**c**) Solid state FT-IR spectra of compound **1** (black), β-CD (blue), and xerogel obtained from compound **1** and β-CD supramolecular gel in DMSO-H$_2$O (red). (**d**) Part of the $^1$H-NMR spectra of compound **1** (red); compound **1** and β-CD Gel in DMSO-D6 (blue), showing shifting of compound **1** aromatic peaks with the addition of β-CD.

Binding stoichiometry is crucial in the inclusion of complex formation. To calculate the binding stoichiometry, bind fit methods were adopted from http://supramolecular. org/software(accessed on 21 February 2023) [36,37]. The emission data at 348, 327, and 408 nm exhibit the best fit with 1:1 stoichiometry (Figure 7b). From Figure 7b, the binding constant between compound **1** and β-CD is 3.77 × 10$^3$ M$^{-1}$ (±4%).

Solid-state FT-IR spectroscopy was used to gain insights intothe interactions in the microrods. In FT-IR, the region of 3500−3200 cm$^{-1}$ is vital for the N−H stretching vibrations; however, the range 1800−1500 cm$^{-1}$ is assigned for the stretching band of amide I and the bending peak of amide II [38]. The FT-IR spectra (Figure 7c, black) of compound **1** exhibit N–H stretching frequency at 3285 cm$^{-1}$ and is responsible for hydrogen-bonded N-H stretching. The ester and amide peaks appeared at 1749, 1638, and 1548 cm$^{-1}$, respectively (Figure 7c, black). The FT-IR spectra of xerogel and β-CD matched quite well (Figure 7c, blue). The FT-IR spectra of xerogel show a broad signal with a peak at around 3285 cm$^{-1}$ assigned as hydrogen-bonded OH (Figure 7c, red). The C-H of the alkyl group appears at 2882 cm$^{-1}$. The carbonyl peak appears at 1638 cm$^{-1}$ (Figure 7c, red). The broad peak at 1060 cm$^{-1}$ is responsible for secondary cyclic alcohols (Figure 7c, red).

Previously, Kakuta and co-workers performed the NMR measurements of gels and demonstratedcomplex formation between CD and guest in gels [39]. Therefore, we conducted the $^1$H-NMR experiments with compound **1** and β-CD in DMSO-D$_6$. From $^1$H-NMR spectra, β-CD interacts with compound **1**. As a result, the aromatic peaks shifted (Figure 7d). However, the H's of the aliphatic group has no shift, indicating the aliphatic group is free (Supplementary Materials Figure S2).

Further, we examined the responsiveness of compound **1** and β-CD supramolecular gel in DMSO-H$_2$O to different stimuli. Due to the formation of the supramolecular polymer

through host–guest interaction [40–44].with compound **1** and β-CD, the gel is thermo-responsive. Upon heating, the supramolecular polymer gel converted to sol and the gel−sol transition temperature was 70 °C (Figure 8). We also tested the gel with picric acid (TNP). The hydrophobic cavity of β-CD cannot accommodate picric acid as a guest molecule [42–44]. Moreover, picric acid can form a charge transfer complex with the aromatic side chain of compound **1**. With the addition of 40 μL of picric acid, the supramolecular polymer gel degraded (Figure 8), but the expected color change due to the formation of a charge transfer complex was not observed. This indicates that the hydrophobic aromatic chain is inside the β-CD cavity and not available from the charge transfer complex with picric acid. Further, we tested the acid responsiveness of the gel. Compound **1** and β-CD supramolecular gel in DMSO-H$_2$O are also responsive to hydrochloric acid. With the addition of 40 μL of 2N HCl, the supramolecular polymer gel converted to sol (Figure 8).

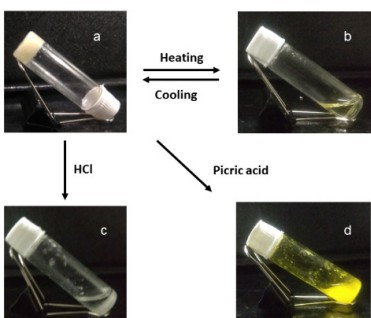

**Figure 8.** The images showing compound **1** and β-CD supramolecular gel in DMSO-H$_2$O is (**a**,**b**) thermo-responsive; (**c**) HCl responsive; and (**d**) picric acid-responsive.

Furthermore, we studied the effect of picric acid on compound 1 and β-CD gel in DMSO by UV-Vis spectroscopy (Supplementary Materials Figure S3) and fluorescence spectroscopy. Figure 9 shows thatcompound **1** exhibits emission at 308 nm. The picric acid exhibits emission bands at 314 and 415 nm. Uponthe formation of compound 1-picric acid charge transfer complex, the peak at 312 nm was quenched and a new peak appeared at 470 nm (Figure 9). However, the emission spectra of compound **1** and β-CD supramolecular gel in DMSO-H$_2$O degraded by picric acid show the characteristic peaks at 312 nm and 410 nm only without any change. This indicates the aromatic group is not free in a supramolecular polymer. This was further supported by fluorescence microscopy studies (Supplementary Materials Figure S3). The quenching of emission due to charge transfer complex formation between compound **1** and picric acid was observed. However, compound **1** and β-CD supramolecular gel in DMSO-H$_2$O degraded by picric acid shows no fluorescence quenching (Supplementary Materials Figure S4).

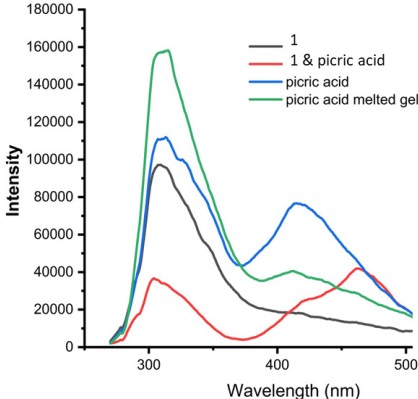

**Figure 9.** The fluorescence spectroscopic studies of compound **1** (black), picric acid (blue), compound 1-picric acid charge transfer complex (red), and compound **1** and β-CD supramolecular gel in DMSO-H$_2$O degraded by picric acid (green). (Excitation at 280 nm).

From the above results, we have proposed a model (Figure 10) for the formation of supramolecular polymer gel because of the host–guest complex between compound **1** and β-CD as well as the decomposition of the supramolecular polymer by the addition of acids.

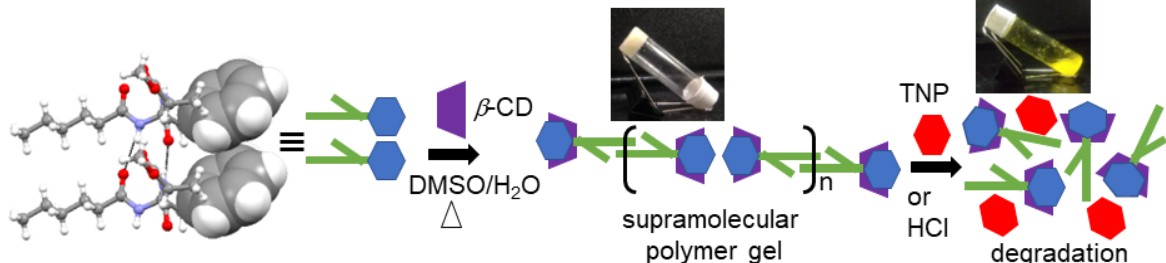

**Figure 10.** The proposed model forsupramolecular polymer formation due tothe inclusion complex between compound **1** and β-CD (**left**) and the decomposition of the supramolecular polymer by addition of picric acid (**right**).

## 4. Conclusions

In conclusion, we report a stimuli-responsive supramolecular polymer gel developed by an amphiphile and β-cyclodextrin (β-CD) in DMSO/water (1:2). From X-ray single crystal diffraction, the amphiphile adopts a kink-like conformation. The amphiphile self-assembled to form a parallel sheet-like structure stabilized by multiple intermolecular hydrogen bonds. Moreover, the parallel sheet-like structure is also stabilized by edge-to-edge π–π stacking interactions. In higher-order packing, it forms a corrugated sheet-like structure stabilized by dispersion forces.The dipeptide amphiphile interacts with β-cyclodextrin and forms gel through supramolecular polymer formation in (DMSO)/water (1:2) by a simple heating-cooling cycle. The sol-to-gel transformation is because of the host–guest complex between compound **1** and β-CD and the formation of supramolecular polymer accompanied by microstructure changes from nanofibers to microrods. The gel is temperature responsive and the $T_{gel}$ is 70 °C. The supramolecular polymer gel is also responsive to stimuli such as picric acid and HCl. The extensive spectroscopic studies show that the aromatic hydrophobic side chain of amphiphile forms a host–guest complex with β-CD. These results will be helpful for the development of programmable responsive materials.

**Supplementary Materials:** The following supporting information can be downloaded at: https://www.mdpi.com/article/10.3390/chemistry5010048/s1, Table S1. Crystal data and structure refinement for compound 1. Figure S1. Images of (a) Compound **1** and α-CD in DMSO-H$_2$O showing no formation of gel; (b) Compound **1** and γ-CD in DMSO-H2O showing no formation of gel.; Figure S2. Part of the $^1$H-NMR spectra of compound **1** and β-CD Gel in DMSO-D$_6$, showing no change of compound 1 aliphatic peaks with addition of β-CD.; Figure S3. Absorption spectra of compound **1** (green), picric acid (red), compound **1**-picric acid (black), picric acid melted gel of compound **1** and β-CD in DMSO-H$_2$O; Figure S4. Fluorescence microscopy images of Compound **1** under (a) white light (b) greenlight, Picric acid under (c) white light (d) green light, Compound **1** + Picric acid under (e) white light (f) green light, Compound **1** + β-CD under (g) white light (h) green light, Picric acid melted Compound **1** + β-CD gel under (i) white light (j) green light.; Figure S5. 1H NMR (500MHz, DMSO-$d_6$, δ in ppm, 298K) spectra of compound **1**; Figure S6. **13C** NMR (125 MHz, DMSO-$d_6$, δ in ppm, 298K) spectra of compound **1**; Figure S7. Mass spectra of Compound **1**; Figure S8. FT-IR spectra of compound **1**. Electronic Supplementary Information available: Supplementary Materials Figures S1–S8, Synthesis, and characterizations.

**Author Contributions:** S.R.C. synthesized the compound. M.D. performed the experimental works. S.S. and M.D. analyzed the data. O.J.I. and D.H. conducted the analysis and wrote the manuscript. All authors have read and agreed to the published version of the manuscript.

**Funding:** This research was funded by CSIR, India (Project No. 02/(0404)/21/EMR-II).

**Institutional Review Board Statement:** Not applicable.

**Informed Consent Statement:** Not applicable.

**Data Availability Statement:** Not applicable.

**Acknowledgments:** M. Douzapau, S. Roy Chowdhury, and S. Singh acknowledge CSIR, India for the research fellowship. We thank the Indian Institute of Science Education and Research Kolkata, India, for its analytical facilities.

**Conflicts of Interest:** The authors declare no conflict of interest.

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
