# Peer review of "Stimuli-Responsive Designer Supramolecular Polymer Gel"

_chemistry, doi:10.3390/chemistry5010048_

Round 1

Reviewer 1 Report

This manuscript by Douzapau et al. describes the preparation and characterisation of an amphiphilic dipeptide derivative, and its behaviour in cyclodextran complexes.  In my opinion, the work is interesting and merits publication.  There are a number of issues with the present manuscript, however, that should first be addressed.

1) It appears that the main subject of the research (compound 1) and its precursors are numbered in the experimental section.  I suggest it would be useful to include a reaction scheme showing the relationship between the reagents used and these various compounds.

2) Could the authors provide a  brief description of the conditions used for the 'protection' reactions with thionyl chloride and methanol (e.g. as mentioned on line 71).  Also, how were by-products - such as Ala-Ala or Phe-Phe - avoided?  Can the authors comment, please?

3) Please define the abbreviations CD (cyclodextrin), DCM (dichloromethane?), DCC and HOBt (butan-1-ol?).

4) Please provide more details of the analytical methods used (e.g. numbers of scans, ESI conditions, step and counting times used for X-ray diffraction etc).

5) I suggest it would be helpful to indicate which bonds correspond to the angles, in Fig. 2, please?

6) The authors describe the crystal structure (Fig. 4) being stabilised by 'hydrophobic interactions' between caproic acid alkyl chains.  As there is no water within the crystal structure, however, 'hydrophobic interactions' is something of a misnomer.  I suggest 'dispersion forces between caproic acid alkyl chains' would be a more accurate description.

7) (L207-208) The authors should specify G' > G" at 0.1% strain amplitude in the frequency sweep measurements.  (Clearly that did not hold generally, as the strain amplitude was increased - in Fig. 5b.)

8) Fig. 6: Please provide scale bars of a suitable size so they can be seen, without having to 'zoom in'.  

9) Also, there appears to be an inconsistency between Figs. 6a and 6b.   Athough (b) appears to be a magnified part of (a), their scale bars appear identical.

Additionally, there were many instances of typographical errors that should be addressed.  Some examples are listed below (with my suggested corrections in parenthesis).

L48: '...markedly slow solvation dynamics in ß-CD than those for water...' (...markedly slower solvation dynamics in ß-CD than those for water...).

L56-57: '...The amphiphile interacts with ß-cyclodextrin and form gel through supramolecular polymer formation...' (...The amphiphile interacts with ß-cyclodextrin and forms gel through supramolecular polymer formation...).

L63: '...potential as eco-friendly advance functional materials...' (...potential as  eco-friendly advanced functional materials...).

L176-177: '....structure has stabilized by...' (....structure was stabilized by...).

L195 '...by inverted vial method...' (...by the inverted vial method...).

L274: '...to from charge transfer complex...' (...to form the charge transfer complex...).

L288: '...picric acid has observed...' (...picric acid was observed...).

L311: '...and form gel...' (...and forms gel...).

L312: '...by simple heating-cooling...' (...by a simple heating-cooling...).

L312-313: '...is because of host-guest complex...' (...is because of the host-guest complex...).

Author Response

1) In response to the Editorial comment, "We check the number of the words in the maintext of your manuscript is 3784, and we suggest you to further enrich the content of the article during the revision.”

Author response

We thank the Editor for reviewing the manuscript and valuable comments.  We have enriched the content accordingly and now the words in the main text are more then 4000.   

2) In response to the comment of reviewer 1, “ This manuscript by Douzapau et al. describes the preparation and characterisation of an amphiphilic dipeptide derivative, and its behaviour in cyclodextran complexes.  In my opinion, the work is interesting and merits publication.  There are a number of issues with the present manuscript, however, that should first be addressed.”

Author response

We thank the reviewer for reviewing the manuscript and appreciation of the research work. We have revised the manuscript accordingly.

  3) In response to the comment of reviewer 1, “1) It appears that the main subject of the research (compound 1) and its precursors are numbered in the experimental section.  I suggest it would be useful to include a reaction scheme showing the relationship between the reagents used and these various compounds.”

Author response

We are sorry for the inconvenience. As suggested, we have incorporated the reaction Scheme in the experimental section. We have revised the manuscript accordingly.

4) In response to the comment of reviewer 1, “2) Could the authors provide a  brief description of the conditions used for the 'protection' reactions with thionyl chloride and methanol (e.g. as mentioned on line 71).  Also, how were by-products - such as Ala-Ala or Phe-Phe - avoided?  Can the authors comment, please?”

Author response

We agree with the reviewer. As suggested we have incorporated the brief description of the conditions used for the 'protection' reactions with thionyl chloride and methanol. To avoid the synthesis of by-products- - such as Ala-Ala or Phe-Phe, we have protected the N-terminus of the D-Phe with caproic acid and C-terminus of the L-Ala with methyl ester. Now coupling between this two using DCC/ HOBT as coupling reagent produced only compound 1. Purification was done by silica gel column (100-120 mesh size) with an ethyl acetate and hexane mixture 1: 9 as the eluent. The product was characterized by 1H NMR, 13C NMR, FT-IR spectroscopy and Mass spectrometry.

5) In response to the comment of reviewer 1, “3) Please define the abbreviations CD (cyclodextrin), DCM (dichloromethane?), DCC and HOBt (butan-1-ol?).”

Author response

We are sorry for the inconvenience. We have defined the abbreviations and revised the manuscript accordingly.  

6) In response to the comment of reviewer 1, “4) Please provide more details of the analytical methods used (e.g. numbers of scans, ESI conditions, step and counting times used for X-ray diffraction etc).”

Author response

As per suggestion, we have provided more analytical details and revised the manuscript accordingly.

7) In response to the comment of reviewer 1, “5) I suggest it would be helpful to indicate which bonds correspond to the angles, in Fig. 2, please?”

Author response

As suggested, we have mentioned which bonds correspond to the angles, in Figure 2 and revised the manuscript accordingly.

8) In response to the comment of reviewer 1, “6) The authors describe the crystal structure (Fig. 4) being stabilised by 'hydrophobic interactions' between caproic acid alkyl chains.  As there is no water within the crystal structure, however, 'hydrophobic interactions' is something of a misnomer.  I suggest 'dispersion forces between caproic acid alkyl chains' would be a more accurate description.”

Author response

We agree with the reviewer and  revised the manuscript accordingly.

9) In response to the comment of reviewer 1, “7) (L207-208) The authors should specify G' > G" at 0.1% strain amplitude in the frequency sweep measurements.  (Clearly that did not hold generally, as the strain amplitude was increased - in Fig. 5b.)”

Author response

As suggested, we have revised the manuscript accordingly.

10) In response to the comment of reviewer 1, “8) Fig. 6: Please provide scale bars of a suitable size so they can be seen, without having to 'zoom in'.  ”

Author response

We are sorry for the inconvenience. As suggested, we have provided the scale bars in Figure 6.

11) In response to the comment of reviewer 1, “9) Also, there appears to be an inconsistency between Figs. 6a and 6b.   Athough (b) appears to be a magnified part of (a), their scale bars appear identical.”

Author response

We are sorry for the misunderstanding. (b) is not a magnified part of (a). (a) is the FE-SEM image of only compound 1 in DMSO solution (self-assembled compound 1). However, (b) is the FE-SEM image of xerogel form by compound 1, β-CD in DMSO and water mixture (supramolecular polymer).

12) In response to the comment of reviewer 1, “Additionally, there were many instances of typographical errors that should be addressed.  Some examples are listed below (with my suggested corrections in parenthesis).

L48: '...markedly slow solvation dynamics in ß-CD than those for water...' (...markedly slower solvation dynamics in ß-CD than those for water...).

L56-57: '...The amphiphile interacts with ß-cyclodextrin and form gel through supramolecular polymer formation...' (...The amphiphile interacts with ß-cyclodextrin and forms gel through supramolecular polymer formation...).

L63: '...potential as eco-friendly advance functional materials...' (...potential as  eco-friendly advanced functional materials...).

L176-177: '....structure has stabilized by...' (....structure was stabilized by...).

L195 '...by inverted vial method...' (...by the inverted vial method...).

L274: '...to from charge transfer complex...' (...to form the charge transfer complex...).

L288: '...picric acid has observed...' (...picric acid was observed...).

L311: '...and form gel...' (...and forms gel...).

L312: '...by simple heating-cooling...' (...by a simple heating-cooling...).

L312-313: '...is because of host-guest complex...' (...is because of the host-guest complex...).”

Author response

We are sorry for the typos.  We have revised the manuscript accordingly.

Reviewer 2 Report

Thank you for sending this work entitled “Stimuli responsive designer supramolecular polymer gel”. I recommend this paper for publication after minor revision which should be corrected before publishing this manuscript.

 1-    There are some grammatical and spelling mistakes. These mistakes should be revised.

2-    Figure 7. A better demonstration is needed by adding the symbols and concentrations for easier follow-up (like Fig. 9).

3-    Page 8-line 274 .. you mean picric acid or hydrochloric acid.

 4-    There is an abbreviation that should be defined or you can use picric acid

Figure 10… TNP

Author Response

13) In response to the comment of reviewer 2, “Thank you for sending this work entitled “Stimuli responsive designer supramolecular polymer gel”. I recommend this paper for publication after minor revision which should be corrected before publishing this manuscript.”

Author response

We thank the reviewer for reviewing the manuscript and appreciation of the research work. We have revised the manuscript accordingly.

14) In response to the comment of reviewer 2, “1-    There are some grammatical and spelling mistakes. These mistakes should be revised.”

Author response

We are sorry for the mistakes.  We have corrected the grammatical and spelling mistakes and revised the manuscript accordingly.

15) In response to the comment of reviewer 2, “Figure 7. A better demonstration is needed by adding the symbols and concentrations for easier follow-up (like Fig. 9).”

Author response

We agree with the reviewer.  We have revised the Figure 7 caption and manuscript accordingly.

16) In response to the comment of reviewer 2, “Page 8-line 274 .. you mean picric acid or hydrochloric acid.”

Author response

We are sorry for the misunderstanding.  It is picric acid. We have revised the manuscript accordingly.

17) In response to the comment of reviewer 2, “There is an abbreviation that should be defined or you can use picric acid

Figure 10… TNP

Author response

We are sorry for the mistakes.  We have defined picric acid (TNP) in page 9, 1st paragraph and revised the manuscript accordingly.